# Serine ADP-ribosylation reversal by the hydrolase ARH3

**Pietro Fontana[1†], Juan José Bonfiglio[2†], Luca Palazzo[1†], Edward Bartlett[1], Ivan Matic[2]\*, Ivan Ahel[1]\***

[1]Sir William Dunn School of Pathology, University of Oxford, Oxford, United Kingdom; [2]Max Planck Institute for Biology of Ageing, Cologne, Germany

**Abstract** ADP-ribosylation (ADPr) is a posttranslational modification (PTM) of proteins that controls many cellular processes, including DNA repair, transcription, chromatin regulation and mitosis. A number of proteins catalyse the transfer and hydrolysis of ADPr, and also specify how and when the modification is conjugated to the targets. We recently discovered a new form of ADPr that is attached to serine residues in target proteins (Ser-ADPr) and showed that this PTM is specifically made by PARP1/HPF1 and PARP2/HPF1 complexes. In this work, we found by quantitative proteomics that histone Ser-ADPr is reversible in cells during response to DNA damage. By screening for the hydrolase that is responsible for the reversal of Ser-ADPr, we identified ARH3/ADPRHL2 as capable of efficiently and specifically removing Ser-ADPr of histones and other proteins. We further showed that Ser-ADPr is a major PTM in cells after DNA damage and that this signalling is dependent on ARH3.

\*For correspondence: imatic@ age.mpg.de (IM); ivan.ahel@path. ox.ac.uk (IA)

[†]These authors contributed equally to this work

Competing interests: The authors declare that no competing interests exist.

## Introduction

ADP-ribosylation (ADPr) is a chemical modification of macromolecules that regulates a wide variety of cellular processes, such as DNA damage repair, transcription, translation, aging, stress responses, microbial pathogenicity and many others (*Gupte et al., 2017*; *Hottiger, 2015*; *Jankevicius et al., 2016*; *Palazzo et al., 2017*). The modification reaction involves the transfer of the ADP-ribose moiety from NAD$^+$ onto the target molecule and the release of nicotinamide (*Gupte et al., 2017*; *Langelier et al., 2012*; *Schreiber et al., 2006*). There are several ADP-ribosyl transferase (ART) enzyme families that can specifically modify proteins (*Bütepage et al., 2015*; *Hassa et al., 2006*; *Rack et al., 2015*). The most thoroughly characterised class of ARTs is the poly(ADP-ribose) polymerase (PARP) family. PARPs are widespread throughout Eukarya and are also found sporadically in bacteria (*Perina et al., 2014*). The human genome encodes 17 members of the PARP family, which are grouped into five distinct classes according to protein domain architecture and involvement in different cellular processes (*Perina et al., 2014*). The majority of these PARPs can catalyse the addition of only one unit of ADP-ribose (mono-ADPr; MAR; MARylation) onto target proteins (*Bütepage et al., 2015*; *Kleine et al., 2008*; *Vyas et al., 2014*). However, several PARP family members such as PARP1, PARP2, and Tankyrases can generate long chains of the repeating ADP-ribose units on their protein targets (poly-ADPr; PAR; PARylation) (*D'Amours et al., 1999*; *Gibson and Kraus, 2012*). Furthermore, PARP family members have displayed differential activities with respect to the preferred amino acid for ADP-ribose attachment. Several amino acids have been reported as acceptors of PARP-mediated modification; most commonly glutamate, but also aspartate, lysine, arginine and cysteine (*Daniels et al., 2015*; *Gagné et al., 2015*; *Martello et al., 2016*; *Vyas et al., 2014*). Most recently, we identified serine as an acceptor of ADPr modification by PARP1 and PARP2 (*Bonfiglio et al., 2017a*; *Leidecker et al., 2016*). We also discovered that the PARP1/2 interacting protein HPF1 (Histone PARylation Factor 1) is responsible for introducing this shift in amino

**eLife digest** Inside cells, genetic information is stored within molecules of DNA. If any of the DNA becomes damaged, the cell has a suite of proteins that can help to repair the DNA. Many of these proteins act as signals that alert the cell to the presence of damaged DNA. One such signal involves adding a molecule called ADPr onto specific proteins that are near the damaged section of DNA.

There are several enzymes that can attach ADPr molecules to proteins and other enzymes known as ADP-ribosylhydrolases can halt the signal by removing the ADPr molecules. Together, these two groups of enzymes control how strong the ADPr signal is, how long it lasts, and therefore control the DNA repair process.

Proteins are made up of building blocks called amino acids. Previous studies have shown that ADPr molecules can be attached to several different amino acids including glutamate, aspartate and cysteine. Specific ADP-ribosylhydrolase enzymes are known to be responsible for removing ADPr molecules from these amino acids. In 2016, a group of researchers found that ADPr can also be added to an amino acid called serine. However, it is not known if cells are able to remove ADPr molecules from serine, or which ADP-ribosylhydrolases might be involved.

Fontana, Bonfiglio, Palazzo et al. – including some of the researchers involved in the earlier work – used biochemical techniques to investigate if any human enzymes are able to remove ADPr molecules that have been attached to serines on proteins. The experiments reveal that the serine ADPr signal increases after DNA damage, before reducing over time. However, in human cancer cells that lack an ADP-ribosylhydrolase known as ARH3, the serine ADPr signal persists after DNA damage. This suggests that adding ADPr molecules to the amino acid serine is a key signal that controls DNA repair and that ARH3 is the main enzyme responsible for erasing this signal.

Drugs that inhibit some of the enzymes that attach ADPr molecules to proteins are used to treat some breast, ovarian and prostate cancers. Therefore, understanding how cells remove these signals from proteins may aid the development of new therapies for these conditions. The next steps following on from this work are to find out more about the structure of ARH3 and to understand how cells that lack this enzyme behave.

acid specificity (*Bonfiglio et al., 2017a*; *Gibbs-Seymour et al., 2016*). Further analyses, by utilising mass spectrometry of human cells treated with DNA damaging agents, revealed large numbers of serine ADP-ribosylation (Ser-ADPr) modifications across a variety of human proteins involved in regulation of genome stability, including histones and PARP1 itself (*Bonfiglio et al., 2017a*; *Bilan et al., 2017*).

Protein ADP-ribosylation can affect structure, function and stability of target proteins, and hence, requires strict control. One mode of regulation is the degradation and/or removal of the ADPr signal by specific enzymes (*Feijs et al., 2013*; *Palazzo et al., 2016*, *2015*; *Rack et al., 2016*). PAR glycohydrolase (PARG) is the most well characterized enzyme in humans for PAR hydrolysis, which utilises a macrodomain fold to bind ADPr and specifically cleaves the ribose–ribose bonds between the subunits of the PAR chains (*Lin et al., 1997*; *Slade et al., 2011*). ADP-ribosylhydrolase 3 (ARH3, also called ADPRHL2) is also able to degrade PAR chains on proteins, but features a different structural composition to PARG, and hydrolyses PAR less efficiently (*Hatakeyama et al., 1986*; *Mueller-Dieckmann et al., 2006*; *Oka et al., 2006*). However, neither PARG nor ARH3 were previously shown to be able to process a single unit of ADPr (mono-ADPr) attached to a protein (*Oka et al., 2006*; *Slade et al., 2011*). This MARylation is instead regulated by a different suite of proteins that do not degrade PAR chains, but rather specifically cleave amino acid-linked ADPr moieties; terminal ADPr protein glycohydrolase (TARG1), MACROD1, MACROD2 and ARH1 (*Barkauskaite et al., 2015*; *Feijs et al., 2013*; *Jankevicius et al., 2013*; *Mashimo et al., 2014*; *Rosenthal et al., 2013*; *Sharifi et al., 2013*). TARG1, MACROD1 and MACROD2 have been shown to hydrolyse acidic residue (Asp/Glu) linked mono-ADPr, whilst ARH1 is capable of removing ADPr from arginine (*Glowacki et al., 2002*; *Jankevicius et al., 2013*; *Rosenthal et al., 2013*; *Sharifi et al., 2013*).

Our recent discoveries that HPF1 enables PARP1-mediated Ser-ADPr suggest a new mechanism of ADPr signalling that would potentially require precise regulation. Additionally, understanding the regulation of Ser-ADPr of histones could provide significant insights to the already complex tapestry of histone modification. Here we endeavoured to determine if Ser-ADPr is a reversible modification and if so, which hydrolase is responsible for the release of this form of ADPr. Our investigations show that DNA damage induced Ser-ADPr modification of histones is reversible in human cells, and that ARH3 is a hydrolase capable of efficiently removing Ser-ADPr from proteins. Here we also establish ARH3 as a much-needed experimental tool to expand the investigations of Ser-ADPr dynamics as our understanding of this recently unveiled form of ADPr (*Leidecker et al., 2016*) is still in its infancy.

## Results

### Histone serine ADP-ribose modification is reversible

PARP-dependent ADP-ribosylation signalling is a rapid and dynamic process in human cells. Within minutes of incurring DNA damage, the peak PARylation signal is observed, before it quickly diminishes (*Figure 1A*) (*Hakmé et al., 2008*; *Mortusewicz et al., 2007*; *Tallis et al., 2014*). Our recent studies have shown that Ser-ADPr is catalysed by PARP1 and PARP2 in a HPF1-dependent manner (*Bonfiglio et al., 2017a*; *Gibbs-Seymour et al., 2016*), and we identified a large number of Ser-ADPr target proteins involved in response to DNA damage. This raised the question of whether Ser-ADPr was a reversible modification as seen for other forms of ADPr and for virtually all the PTMs playing a role in cell signalling. To investigate this, we carried out quantitative proteomics experiments using stable isotope labeling by amino acids in cell culture (SILAC) (*Ong et al., 2002*) in combination with our recently developed mass spectrometric approach (*Bonfiglio et al., 2017a*; *Leidecker et al., 2016*). U2OS heavy labeled (Lys8) cells were treated with $H_2O_2$ at different time points (0 min, 10 min and 120 min) and then combined with light labelled $H_2O_2$-tretead cells (Lys0) that were used as a spike-in SILAC standard (*Figure 1B*) (*Geiger et al., 2011*). As expected, all the detected Ser-ADPr marks increased in cells exposed to oxidative DNA damage for 10 min (*Figure 1C–D* and *Figure 1—figure supplement 1*, *Figure 1—source data 1*) (*Leidecker et al., 2016*). However, after 120 min of $H_2O_2$ treatment, Ser-ADPr returned to levels similar to the untreated cells (*Figure 1C–D* and *Figure 1—figure supplement 1*, *Figure 1—source data 1*), demonstrating that Ser-ADPr is also a reversible modification.

### ARH3 hydrolyses Ser-ADPr

As shown above, histone Ser-ADPr marks are reversed at a late time point after DNA damage, roughly mirroring the global PARylation and total ADPr profiles (*Figure 1*). This raised the question of how ADPr attached to serine residues is removed from its protein targets. We reasoned that a known hydrolase or its homologue could be responsible for the removal of Ser-ADPr. Previous research has found protein ADP-ribosyl hydrolase activity in proteins bearing one of two evolutionarily unrelated domains: a ADP-ribosylglycohydrolase (DraG-like) fold (pfam PF03747) and a macrodomain (pfam PF01661) (*Glowacki et al., 2002*; *Oka et al., 2006*; *Rack et al., 2016*). Therefore, we screened all known human ADP-ribosylhydrolases, belonging to either the macrodomain family (PARG, TARG1, MACROD1, and MACROD2) or to DraG family (ARH1, ARH2 and ARH3) for their ability to cleave Ser-ADPr. We also tested human macrodomains (such as ALC1 and MACROH2A) that are thought to be devoid of catalytic activity (*Ahel et al., 2009*; *Timinszky et al., 2009*). We used a well-defined histone H3 peptide fragment (aa 1-21) as a known substrate for serine mono-ADP-ribosylation (*Bonfiglio et al., 2017a*) which we then incubated with each of the previously mentioned proteins (*Figure 2A*). Strikingly, only ARH3 was able to catalyse removal of ADPr from the histone peptide, implicating a serine ADP-ribosylhydrolase activity. Preceding studies with ARH3 found poly(ADP-ribosyl) hydrolase activity, but inability to remove terminal arginine, or other amino acid, ADP-ribose linkages that were known prior to discovery of Ser-ADPr (*Oka et al., 2006*). As expected, we also observed that both PARG and ARH3 were able to degrade the PAR chains on automodified PARP1 (*Figure 2A*, lanes 2 and 8), albeit ARH3 was less efficient in removing the chains (*Figure 2B*, *Figure 2—figure supplement 1A*) (*Mueller-Dieckmann et al., 2006*). We then performed a similar experiment using full length recombinant histone H1, which can be ADP-

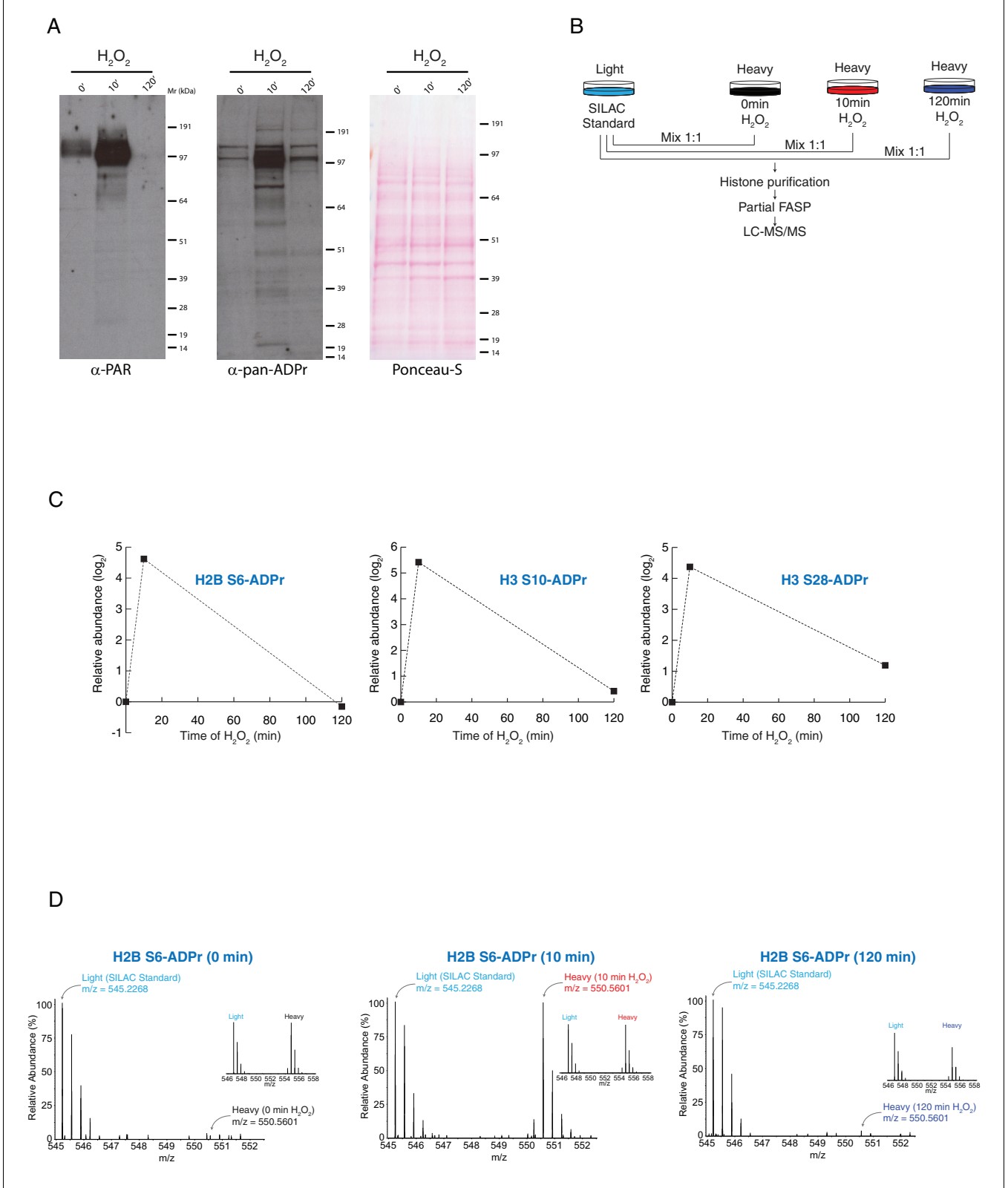

**Figure 1.** Histone serine ADP-ribose modification is reversible. U2OS cells were treated with $H_2O_2$ and analysed at indicated time-points. (**A**) The samples were lysed and the proteins were separated by SDS-PAGE, analysed by western blot and probed for PAR (left) or pan-ADPr (middle). Ponceau-S staining was used as loading control (right). (**B**) Schematic representation of the SILAC-based strategy to quantify core histone Ser-ADPr marks after different time points of 2 mM $H_2O_2$–induced DNA damage. Light labeled (Lys0) cells treated for 10 min with 2 mM $H_2O_2$ were used as
*Figure 1 continued on next page*

*Figure 1 continued*

SILAC Standard. (C) The relative abundance of Ser-ADPr modification on histone proteins H2B (Ser6) and H3 (Ser10, Ser28) was calculated and plotted as a function against time of $H_2O_2$ treatment. (D) MS1s of a Ser-ADPr H2B peptide (H2B Ser6-ADPr) at different time points of 2 mM $H_2O_2$ treatment. The heavy peptide was derived from cells treated with $H_2O_2$ for the indicated time point, and the light peptide was derived from the SILAC Standard (10 min $H_2O_2$-treated cells). Each inset (right) shows a ~1:1 ratio (heavy/light) of a non-ADP-ribosylated peptide from the same experiment.

The following source data and figure supplement are available for figure 1:

**Source data 1.** MaxQuant evidence table related to *Figure 1—figure supplement 1*.

**Figure supplement 1.** Histone serine ADP-ribose modification is reversible.

ribosylated by PARP1 on serine residues in the presence of HPF1 (*Bonfiglio et al., 2017a*). Whilst ARH3 was able to remove both the PARylation and mono(ADP-ribosyl)ation of H1, PARG was only able to degrade the PAR chains on histone H1 and PARP1, leaving a mono(ADP-ribosyl)ation signal that was visible as two distinct bands at the sizes of H1 and PARP1 (*Figure 2B*). This may be expected given the inability of PARG to process mono(ADP-ribosyl)ation (*Slade et al., 2011*). The other glycohydrolases had no discernible effect on the modification of either PARP1 or histone H1. A double digestion with PARG/ARH3 confirmed that most of the ADP-ribosylated chains had been attached to serine residues (*Figure 2—figure supplement 1*).

Next, we contrasted the poly- and mono-ADP-ribosylation of PARP1 and histone H1. We prepared these substrates by incubating H1 and HPF1 together with either wild type, or E988Q mutant PARP1 for a modification reaction. The E988Q mutation alters the activity of PARP1 to that of mono-ADP-ribosylation, so in this way we were able to generate Ser mono-ADPr modifications on a protein target that is otherwise PARylated under wild type conditions (*Figure 2—figure supplement 1B–C*). The activities of the hydrolases tested against PARylation products yielded the same result as before in *Figure 2A* – both PARG and ARH3 degraded the PAR (*Figure 2C*, lanes 2 and 4 - upper panel); and collapsed the protein signal for PARP1 and H1 proteins as seen by Coomassie stain (*Figure 2C*, compare lane 1 with lanes 2 and 4; - lower panel). However, the serine mono-ADP-ribosylated histone H1 samples were only hydrolysed by ARH3 (*Figure 2C*, lane 9), with no notable reduction in mono-ADP-ribose band intensity by the other hydrolases, including PARG. Although it cannot be resolved well in these gels, we consistently observed a small but notable shift in the stained protein signal after the reaction with ARH3, suggesting the reversal of serine mono(ADP-ribosyl)ated H1 into its unmodified form (*Figure 2C* compare lane 9 to lanes 6, 7, 8, and 10).

To investigate whether ARH3 is also capable of removing ADPr from histones purified from cells that had been modified on serine upon DNA damage (*Bonfiglio et al., 2017a*; *Leidecker et al., 2016*), we carried out quantitative proteomics experiments using a SILAC approach. Core histones purified from heavy labeled $H_2O_2$-treated cells (Lys8) were incubated in the presence or absence of ARH3 for 30 min. To quantitatively evaluate the removal of Ser-ADPr by ARH3, core histones purified from light labelled $H_2O_2$-treated cells (Lys0) were added as a spike-in SILAC standard. All the identified histone ADPr peptides were significantly decreased in samples treated with recombinant ARH3 (*Figure 2D*, *Figure 2—source data 1*), demonstrating that this hydrolase is also able to remove Ser-ADPr from endogenous histones.

We have recently characterised the Ser-ADPr sites on the isolated automodifcation domain of PARP1 (aa 374-525) when incubated in the presence of full length PARP1 and HPF1 (*Bonfiglio et al., 2017a*). During this reaction, the isolated domain is modified on 3 serine residues and thus we were able to use it as a well-defined model to analyse removal of the Ser-ADPr from an ADP-ribosylated non-histone protein. We used purified PARP1 automodification domain with HPF1 and either wild type PARP1 or PARP1 E988Q mutant, to produce PARylated and MARylated substrates (respectively) for the hydrolase reactions and then tested them with ARH3, MACROD2, TARG1 or ARH1 (*Figure 2E*). ARH3 was efficient at removing Ser-ADPr modifications from both the mono- and poly-ADP-ribosylated automodification domain, confirming the capability to hydrolyse serine linked ADP-ribose from protein targets besides histones.

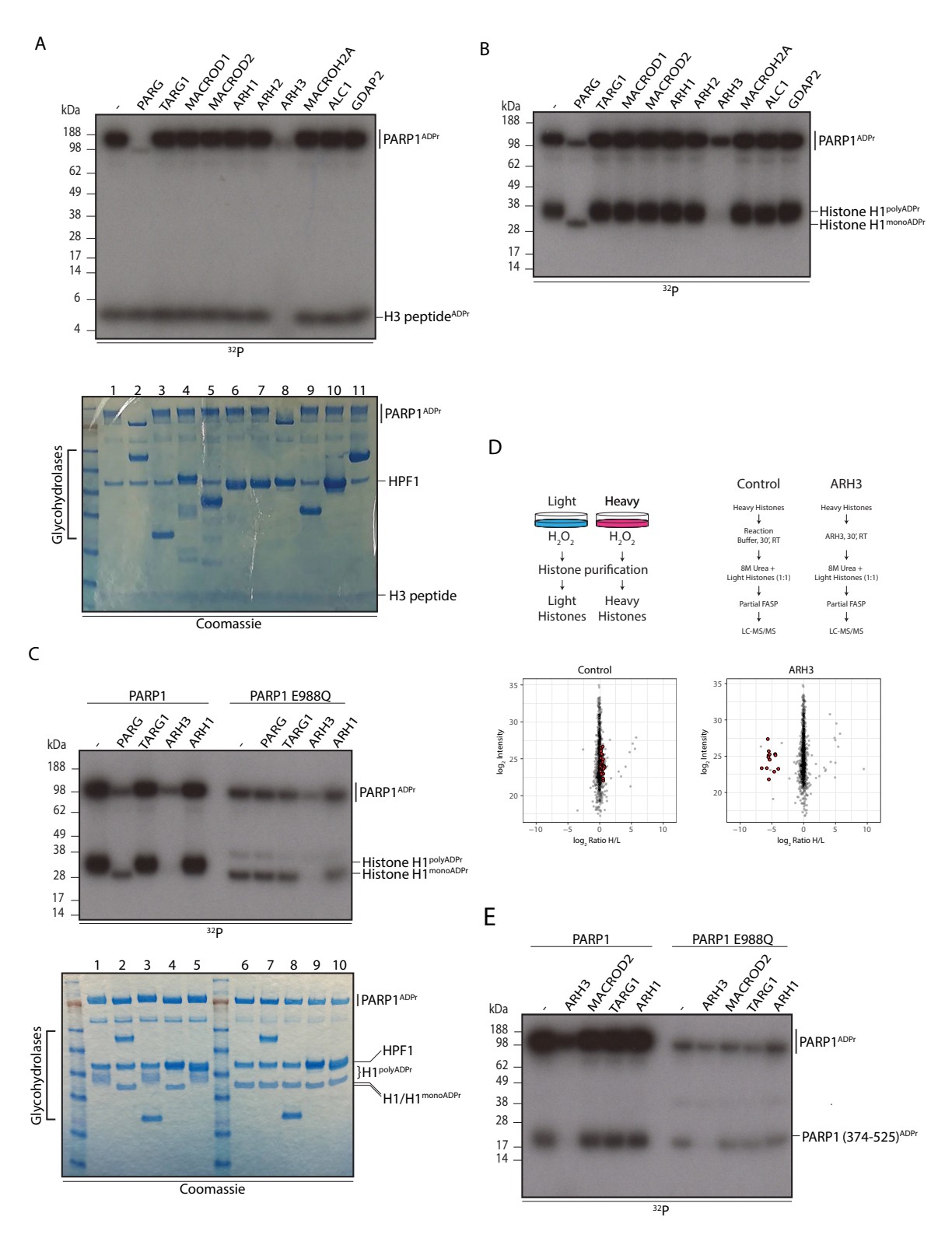

**Figure 2.** ARH3 hydrolyses Ser-ADPr. (**A**) Glycohydrolysis of mono-ADP-ribosylated H3 histone peptide (aa 1-21). ADP-ribosylated H3 peptide was obtained by incubation with PARP1 and HPF1 using radioactively labelled NAD+ as an ADPr donor. After stopping the modification reaction with PARP inhibitor, the indicated glycohydrolases and related proteins were added to the substrate as shown on the figure. Lower panel – Coomassie stained gel of the proteins used in the reaction that also serves as a loading control. (**B**) Removal of poly-ADP-ribosylation from the recombinant histone H1

*Figure 2 continued on next page*

*Figure 2 continued*

substrate by various glycohydrolases. Reaction performed as described in A. (C) Comparison of the hydrolase reactions on the poly- and mono-ADP-ribosylated histone H1 substrates. Lower panel – Coomassie stained gel of the proteins used in the reaction that also serves as a loading control. (D) Schematic representation of the SILAC-based strategy to quantify ADPr removal upon in vitro incubation of purified Ser-ADPr histones with or without recombinant ARH3 (top panels). Log$_2$ of summed peptide intensities were plotted against log$_2$ Heavy/Light SILAC ratios for each condition (bottom panels). ADPr peptides are colored in red. (E) Autoradiogram analysis of Ser mono-ADPr hydrolysis with a non-histone protein substrate. The ADP-ribosylated automodification domain of PARP1 (aa 374-525) was used in this assay and the reactions were supplemented with the indicated hydrolases. In panel panels A, B, C and E, the reaction products were separated by SDS-PAGE and analysed by autoradiography. The signals relating to the specific ADP-ribosylated protein are indicated in each panel.

The following source data and figure supplement are available for figure 2:

**Source data 1.** MaxQuant evidence table related to *Figure 2D*.

**Figure supplement 1.** PARP1 E988Q generates mono Ser-ADPr on itself and on substrates.

## Characterisation of the serine ADP-ribosylhydrolase activities of ARH3

To precisely characterise the activity of ARH3 on Ser-ADPr and to exclude the effect of PARP1 and HPF1 that were present in the previous hydrolase reactions, we modified the H3 histone peptide (aa 1-21) with PARP1 and HPF1 as described above, and then purified the peptide fragment away from proteins for incubation with the same suite of hydrolases as before (*Figure 3A*). The autoradiogram demonstrated that ARH3 alone was sufficient for the complete removal of Ser-ADPr from H3 histone peptide in the absence of PARP1 and HPF1.

Our previous study established that Ser6 of H2B is ADP-ribosylated by PARP1 in the presence of HPF1 (*Bonfiglio et al., 2017a*), and similarly as for H3, we could show that the purified ADP-ribosylated H2B peptide is efficiently and specifically de-ADP-ribosylated by ARH3 (*Figure 3B*).

Earlier research on ARH3 identified the critical acidic amino acids Asp77 and Asp78 that are required for coordination of a magnesium ion and poly(ADP-ribose) hydrolysis (*Oka et al., 2006*) (*Figure 3C*), so we generated single site (D77N and D78N) as well as double site (D77N D78N) ARH3 mutants to confirm whether the same residues were necessary for Ser-ADPr hydrolysis (*Figure 3D and E*). Only wild type ARH3 was able to remove the Ser-ADPr modification from the histone H3 and H2B peptides. These results show that Asp77 and Asp78 are necessary for both PAR and serine-ADPr hydrolysis. Further, in the presence of the metal chelating agent EDTA, ARH3 was unable to hydrolyse the substrate, confirming that the activity is Mg$^{2+}$ dependent (*Figure 3D and E*). Furthermore, we showed that ARH3 enzymatic activity is concentration and time dependent (*Figure 3F and G*). Notably, ARH3 was highly efficient against Ser-ADPr and quantitatively hydrolysed the substrate at nanomolar enzyme concentrations.

## Specificity of ARH3 activity

To assess the amino acid specificity of ARH3, we asked whether ARH3 presents activity on ADP-ribosylations linked to other amino acids in addition to O-linked Ser-ADPr. To explore this hypothesis we performed a PARP1 automodifcation assay with PARP1 E988Q in the absence of HPF1, a reaction that we have previously shown to produce glutamate linked mono-ADPr (*Sharifi et al., 2013*). The mono-ADP-ribosylated PARP1 E988Q protein was then used as a substrate and incubated with ARH3, along with two characterised glutamate mono-ADP-ribose hydrolases; MACROD2 and TARG1, and ARH1 which has been shown to possess arginine mono-ADP-ribose hydrolysis activity (*Jankevicius et al., 2013*; *Oka et al., 2006*; *Sharifi et al., 2013*). The autoradiogram shows that both MACROD2 and TARG1, but not ARH1, nor ARH3, were able to catalyse the efficient removal of glutamate mono-ADP-ribose from the PARP1 protein (*Figure 4A*). We also tested ARH3 against arginine-linked ADPr. We prepared the substrates for this reaction by treating K562 human myelogenous leukemia cell extract with recombinant ARTC2.2 protein, an arginine ADP-ribosyltransferase (*Adriouch et al., 2008*) in the presence of [$^{32}$P]NAD. *Figure 4B* shows that only ARH1 (that was used as a positive control), but not ARH2, ARH3 or TARG1, efficiently cleaved the arginine ADP-ribosylated proteins in the cell extracts.

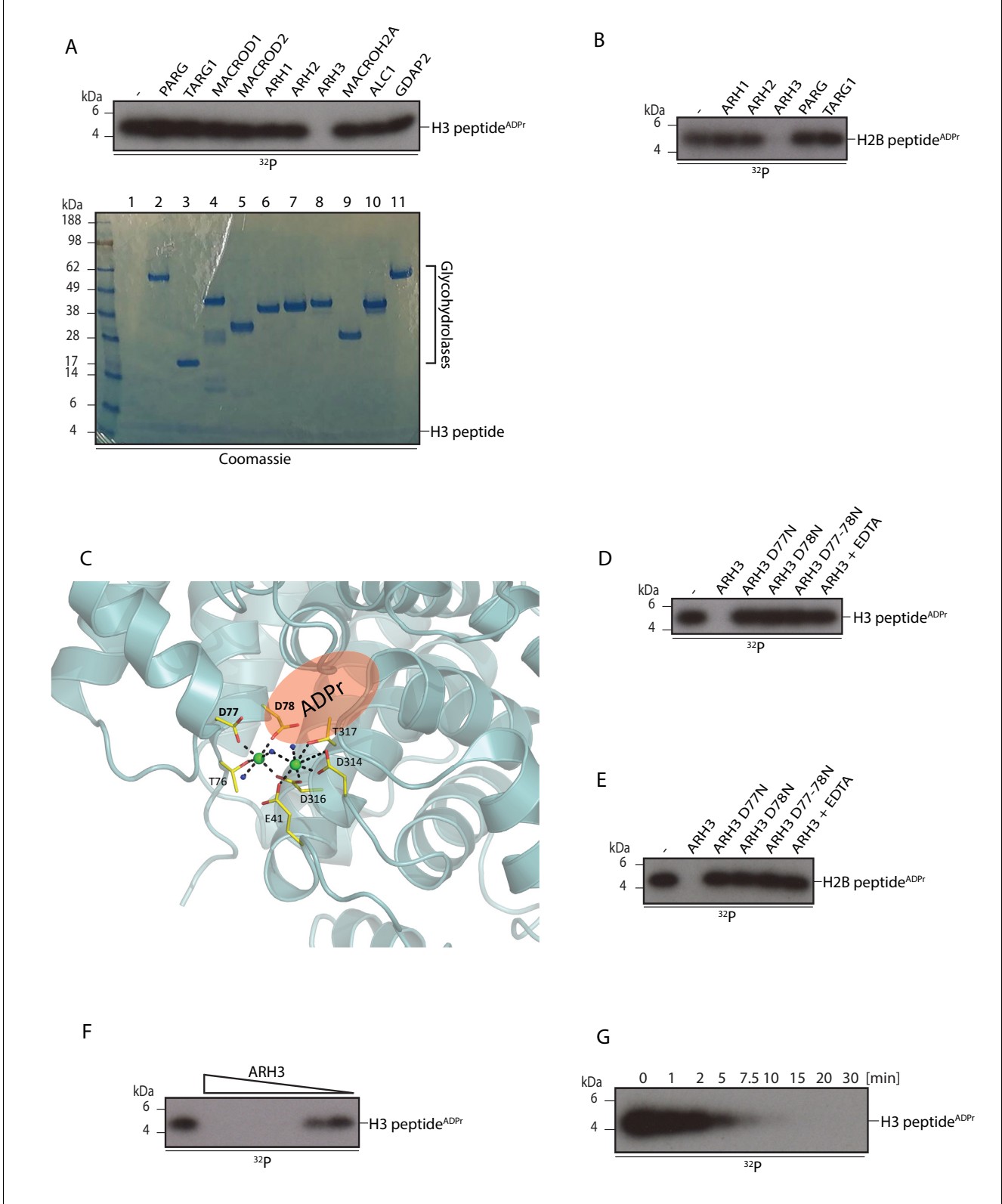

**Figure 3.** Characterisation of the serine ADP-ribosylhydrolase activities of ARH3. Glycohydrolase assay using the purified serine ADP-ribosylated histone peptides as substrates. Reactions were performed and analysed as described in *Figure 2A*. (**A**) Activity of different hydrolases against ADP-ribosylated H3 peptide (aa 1-21). (**B**) The same as in (**A**) except purified histone H2B peptide was used. (**C**) ARH3 structure (PDB ID: 2FOZ) showing the amino acid residues in the catalytic site coordinating Mg ions (green). The putative ADPr binding site is highlighted in red. (**D**) Mutating ARH3 catalytic

*Figure 3 continued on next page*

*Figure 3 continued*
residues or addition of EDTA to the reaction abolishes ARH3 activity. (E) Reaction performed as (D) except histone H2B peptide was used. (F) ARH3 activity is enzyme concentration dependent. ARH3 enzyme concentrations were in the range from 3 μM to 5 nM. (G) ARH3 activity is time dependent. 0.2 μM ARH3 was used in this assay.

We also tested the ARH3 and TARG1 against H3 peptide with chemically generated lysine-linked ADPr (*Jankevicius et al., 2013*) (*Figure 4C*). We observed no activity of ARH3, nor TARG1 against this substrate. All together, these findings confirm that ARH3 is unable to catalyse the efficient removal of glutamate, arginine or lysine linked ADP-ribose moieties, and is specific for Ser-ADPr hydrolysis.

## ARH3 is responsible for removal of serine ADPr in cells

To confirm the relevance and the enzymatic activity of ARH3 in cells, we generated ARH3 knock-out (KO) U2OS cells lines using CRISPR-Cas9 genome editing technology. Control and ARH3 KO cells were treated with 2 mM $H_2O_2$ and harvested at 0, 10 min and 120 min post DNA damage. In order to detect the effects of ARH3 KO on ADPr levels we used two different antibodies; the widely used anti-PAR antibody which recognises only long PAR chains, and a macrodomain recombinant fusion protein that binds all forms of ADPr including mono and short oligo-ADPr (described here as a pan-ADPr antibody) which has been recently developed (*Gibson et al., 2016*) (*Figure 5A*). When compared to control cells, ARH3 KO cells showed higher levels of ADP-ribosylated proteins under unstimulated conditions. As shown in *Figure 5A*, ADP-ribosylation is dramatically induced in both wt and ARH3 KO cells following DNA damage (see 10 min samples), particularly on PARP1 and Histone proteins. However, the difference between wt and ARH3 KO cells became dramatic at 2 hr post DNA damage, in the ARH3 KO cells the ADPr signal persisted whilst in the wild type cells the ADPr signal

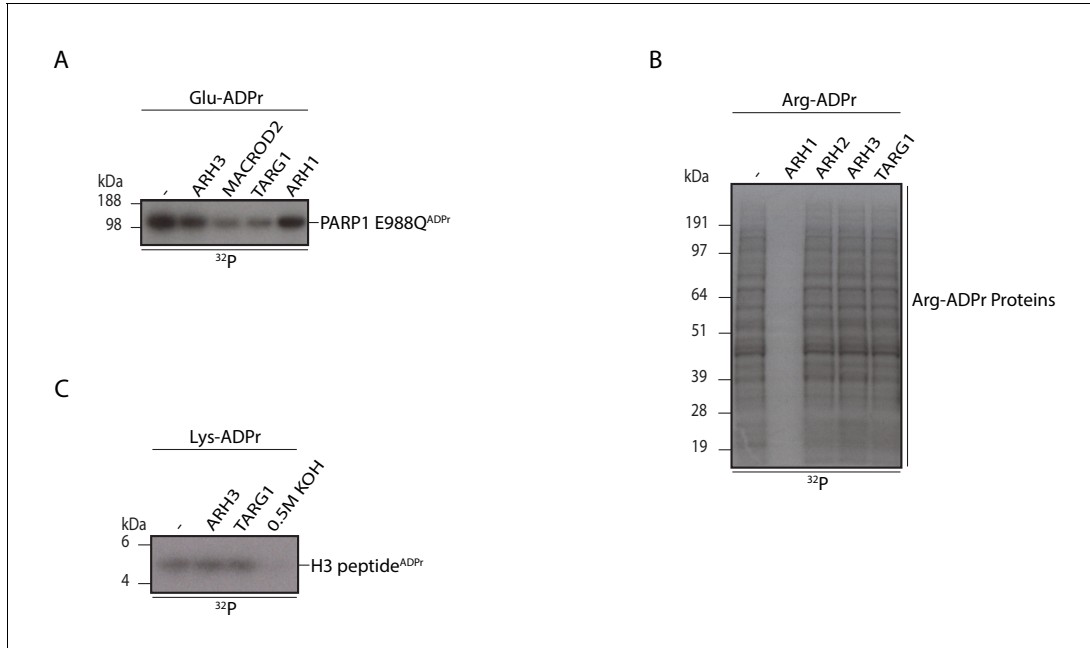

**Figure 4.** ARH3 specifically cleaves Ser mono-ADPr, but not Glu, Arg or Lys mono-ADPr. Reactions were analysed as in *Figure 2A*. (A) Analysis of ARH3 activity on a glutamate-ADP-ribosylated protein. The mono-ADP-ribosylated recombinant PARP1 E988Q protein was treated with the indicated glycohydrolases. (B) Analysis of ARH3 activity on arginine-ADP-ribosylated proteins. Cellular extracts from K562 cells were ADP-ribosylated by recombinant ARTC2.2 protein. The reactions were then supplemented with the indicated glycohydrolases. (C) Analysis of ARH3 activity on a lysine-ADP-ribosylated peptide. A chemically modified histone H3 peptide with mono-ADPr on lysine residues was incubated with the indicated glycohydrolases or 0.5 M KOH for 30 min.

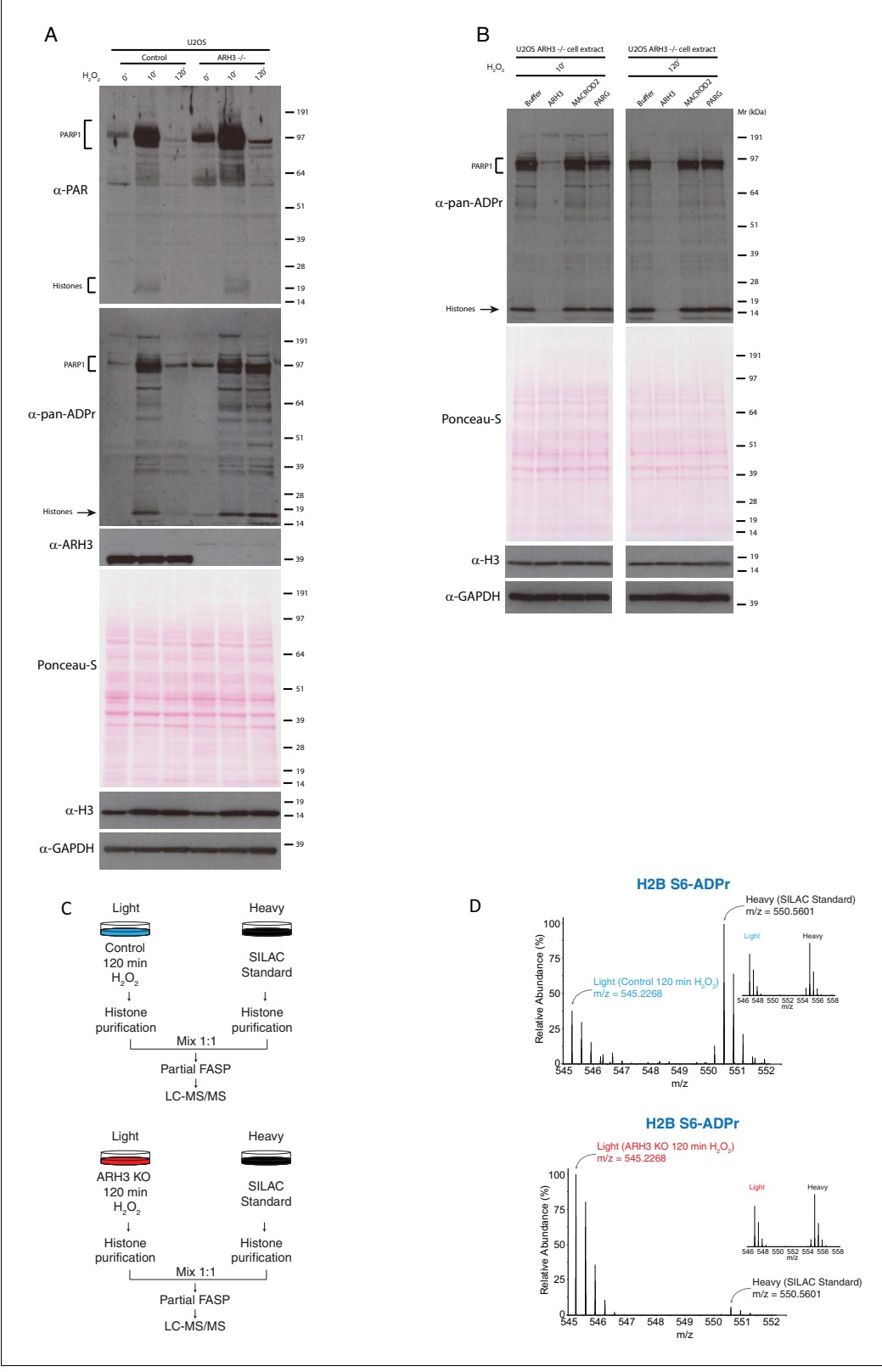

**Figure 5.** ARH3 is necessary for removal of Ser-ADPr in cells. (**A**) Control and ARH3 KO U2OS (ARH3$^{-/-}$) cells were treated with 2 mM H$_2$O$_2$ for the indicated time points. After treatment, cells were lysed and proteins were separated by SDS-PAGE, analysed by western blot and probed for PAR, pan-ADPr, ARH3, H3, and GAPDH antibodies. Additionally, Ponceau-S staining was used as loading control (**B**) Cell extracts obtained from ARH3 KO cells treated with 2 mM H$_2$O$_2$ for 10 and 120 min were incubated with buffer or ADP-ribosylhydrolases ARH3, MACROD2 and PARG. Samples were

*Figure 5 continued on next page*

*Figure 5 continued*

separated by SDS-PAGE, analysed by Western blot and probed for pan-ADPr, H3, GAPDH, and Ponceau-S staining were used as loading control. (C) Schematic representation of the SILAC-based strategy to quantify core histone Ser-ADPr marks from Control U2OS cells (top panel) or ARH3 KO U2OS cells (bottom panel) after $H_2O_2$ treatment for 120 min. Heavy labeled (Lys8) Control U2OS cells treated for 10 min with $H_2O_2$ were used as SILAC Standard. (D) MS1s of a Ser-ADPr H2B peptide (H2B Ser6-ADPr) from Control U2OS cells (top panel) or ARH3 KO U2OS cells (bottom panel) after $H_2O_2$-treatment for 120 min. The light peptide was derived from Control (top panel) or ARH3 KO (bottom panel) cells treated with 2 mM $H_2O_2$ for 120 min, and the heavy peptide was derived from the SILAC Standard (10 min $H_2O_2$ treated cells). Each inset shows a ~1:1 ratio (heavy/light) of a non-ADP-ribosylated peptide from the same experiment. In the ARH3 KO U2OS cells, the chosen H2B mark (H2B S6-ADPr) resulted ~40 times more abundant than in the Control U2OS cells.

The following source data is available for figure 5:

**Source data 1.** MaxQuant ADPr sites table related to *Figure 5C–D*.

was greatly reduced. This observation was most clear with the short oligo- and mono-ADPr as detected by pan-ADPr antibody, while turnover of long chains (see the signal at PARP1 detected by PAR antibody) was much less affected (*Figure 5A*). This may suggest that PARG is required in an early phase of DNA-damage response, shortening long chains of Ser-ADPr substrates, producing mono (and likely very short chains) which subsequently are substrates of ARH3 enzymatic activity.

To further confirm that deficiency of ARH3 is directly responsible for the persistence of the ADPr signal, we supplemented the ARH3 KO extracts (collected post DNA damage) with addition of purified recombinant ARH3. As seen in *Figure 5B*, ARH3 erases the persisting ADPr signal (at 120 min post DNA damage), confirming that ARH3 is both necessary and sufficient for removal of these modifications. MACROD2 and PARG were not capable of hydrolysing the same substrates, when added to the extracts and probed with pan-ADPr, suggesting that accumulation of monoADPr was on serine residues (*Figure 5B*).

To directly confirm that Ser-ADPr was indeed accumulated in these conditions, we analysed by mass spectrometry the isolated histones from these cell extracts. Histones purified from U2OS heavy labelled $H_2O_2$-treated cells (Lys8) were used as a spike-in SILAC standard (*Figure 5C*). As expected, after 120 min of DNA damage, the levels of histone Ser-ADPr were dramatically increased in ARH3 KO cells compared the wild type cells. (*Figure 5D*, *Figure 5—source data 1*).

## ARH3 as a tool for recognizing histone serine ADPr

Mass spectrometry is currently the only available technique for the detection of Ser-ADPr. Given that we have determined that ARH3 is specific for Ser-ADPr and that this hydrolase does not remove ADPr from several other known target amino acids, we set to employ ARH3 for the establishment of a simple alternative approach for monitoring the modification of known Ser-ADPr target proteins. We reasoned that the new pan-ADPr reagent (*Gibson et al., 2016*), which by itself does not distinguish between different amino acid specificities, in combination with ARH3 could discern the presence of histone Ser-ADPr by immunoblotting. Identically to what we observed for in vitro Ser-ADPr on peptides and proteins, ARH3 completely removes ADPr from core histones derived from DNA damage treated cells (*Figure 6A*). This confirms our previous mass spectrometric analyses that conclusively identified histone ADPr on serine, but not other residues (*Leidecker et al., 2016*), thus providing further indication of serines as the exclusive modification sites on core histones under these experimental conditions. Next, we implemented this newly established approach to confirm the reversibility of histone Ser-ADPr in DNA repair indicated by our SILAC spike-in experiment (*Figure 1C,D* and *Figure 1—figure supplement 1*). As expected, following a strong induction of histone Ser-ADPr after 10 min of treatment with peroxide, the modification is completely reversed after 120 min (*Figure 6B*). This strategy based on ARH3 is set to enhance and significantly facilitate any further investigation of Ser-ADPr on established substrates.

## Discussion

PTMs modulate virtually all cellular processes by acting as molecular switches that modify properties of target proteins including function, interactions and stability. As with all homeostatic and stress

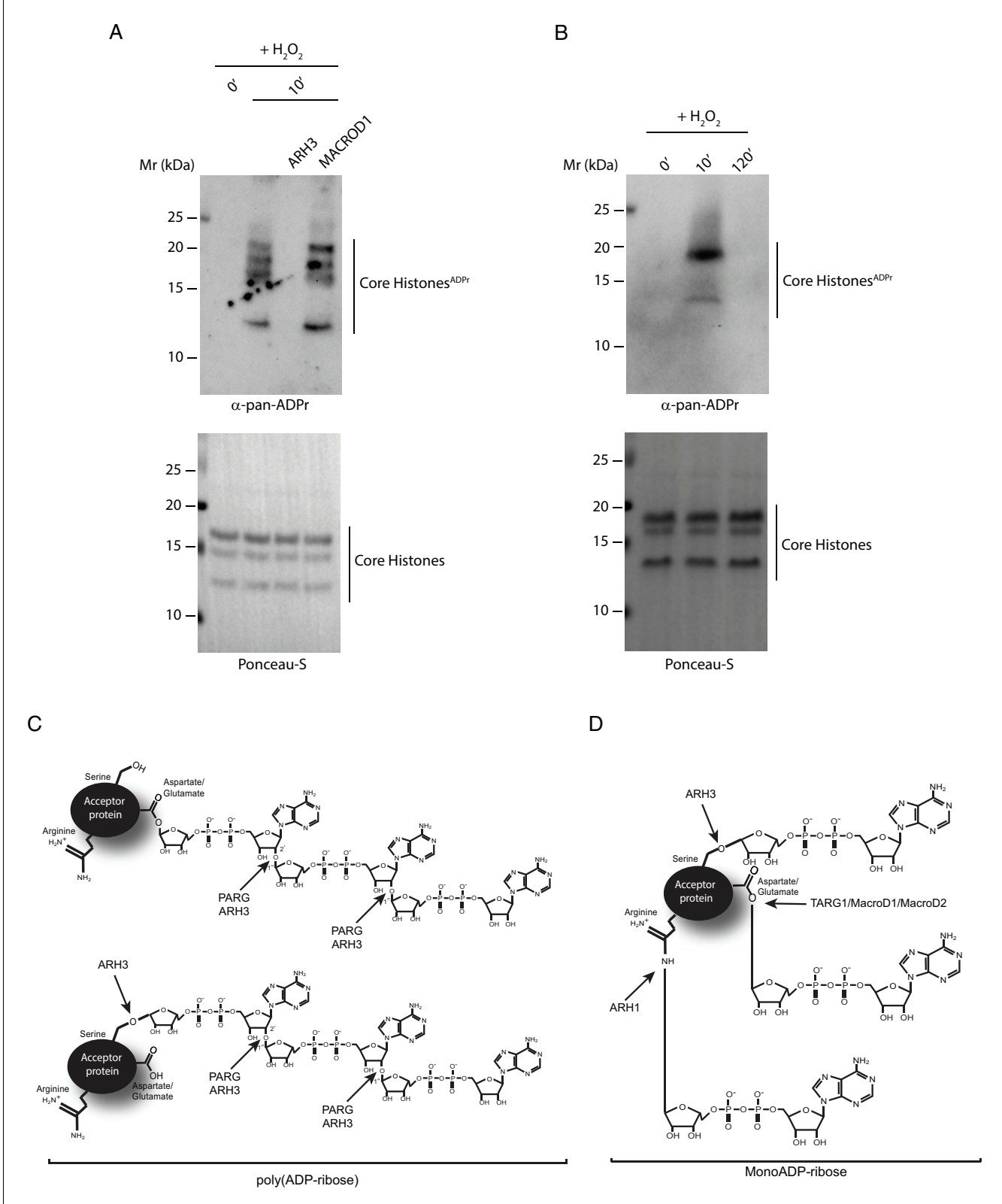

**Figure 6.** ARH3 as a tool for recognizing histone Ser-ADPr. (**A**) Core histones were purified from U2OS cells treated with $H_2O_2$ for the indicated time points. Recombinant ARH3 or MACROD1 or reaction buffer was added to histones purified from 10 min $H_2O_2$-treated cells. After treatment, proteins were separated by SDS-PAGE, analysed by western blot and probed for pan-ADPr. Ponceau-S staining was used as loading control. (**B**) Purified core histones from U2OS cells treated with $H_2O_2$ for the indicated time points were separated by SDS-PAGE, analysed by Western blot and probed for pan-

*Figure 6 continued on next page*

*Figure 6 continued*

ADPr. Ponceau-S staining was used as loading control. (**C** and **D**) A schematic representation of the specificity of the ADP-ribosylhydrolases PARG, ARH3, TARG1, MACROD1, MACROD2 and ARH1 for PARylated (**C**) and MARylated (**D**) proteins.

control systems, this level of regulation relies on reversibility to maintain balance, in this case the fine-tuning of the attachment and removal of PTMs. In fact, disruption of the enzymes responsible for deconjugational (and conjugational) dynamics of PTMs often leads to disease states. This also applies to ADP-ribosylation, as disregulation of mono- or poly-ADPr processes can result in neurodegenerative disorders and cancer (*Bütepage et al., 2015*; *Hanai et al., 2004*; *Rouleau et al., 2010*; *Sharifi et al., 2013*).

We recently discovered serine ADP-ribosylation (Ser-ADPr) as a novel modification of proteins involved in genome stability (*Bonfiglio et al., 2017a*; *Leidecker et al., 2016*). Our data identified nearly 300 hundred Ser-ADPr modification sites in proteins extracted from living cells, suggesting that this PTM is widespread across a breadth of pathways including; DNA repair, chromatin organisation, mitotic nuclear division, DNA recombination, transcriptional regulation and mRNA splicing amongst others (*Bonfiglio et al., 2017a*). We further showed that at least a fraction of these Ser-ADPr targets are modified by the HPF1/PARP1/2 complex, including histones, PARP1 itself and High Mobility Group (HMG) proteins, as we were able to reconstitute the sites in vitro using recombinant proteins (*Bonfiglio et al., 2017a*; *Gibbs-Seymour et al., 2016*). These findings raised an obvious question of whether Ser-ADPr modification is reversible and which enzyme is responsible for said activity. ARH3 has previously been demonstrated to cleave PAR chains, although less rapidly than PARG, but also to hydrolyse O-acetyl-ADP-ribose by-products generated by Sir2 deacetylases (*Mashimo et al., 2013*; *Oka et al., 2006*; *Ono et al., 2006*). Here, we established that ARH3 is also capable of efficiently removing Ser-ADPr in a magnesium-dependent manner, and that mutation of key aspartate residues (which are also critical for PAR cleavage) ablates this activity. Given our previous discoveries that Ser-ADPr is a widespread modification in the human proteome following DNA damage, the identification of ARH3 as the first enzyme that is able to hydrolyse Ser-ADPr may prove a critical advance in understanding control of NAD signalling and ADP-ribosylation in regulation of genome stability (*Bonfiglio et al., 2017a*; *Leidecker et al., 2016*) (*Figure 6C and D*).

Prior observations suggest a wide distribution of ARH3 throughout the cell; cytosol, nucleus and mitochondria—an interesting array of localisations given that the ARH3 protein possesses a mitochondrial targeting peptide at the N-terminus (both in Human and Murine proteins) (*Mashimo et al., 2013*; *Niere et al., 2012*). Despite the wide cellular distribution, all described ARH3 functions so far seem to converge on safeguarding genome stability. Previous studies suggested that ARH3 functioned in the degradation of PAR chains that had been released by both nuclear and cytosolic PARG following the oxidative stress exposure. As a result of this cooperation, ARH3 buffers the Parthanatos response (a Caspase-independent cell death induced by the transport and binding of PAR to mitochondrial protein AIF) (*Mashimo et al., 2013*). Furthermore, ARH3 KO MEFs were found to be sensitive to hydrogen peroxide compared to wild type cells, and in the absence of ARH3 protein PAR was observed to accumulate in the nucleus (*Mashimo et al., 2013*). Additionally, the sensitivity of the ARH3 KO MEFs to hydrogen peroxide was rescued by treatment with a PARP1 inhibitor, linking the function of ARH3 to the PARP1 controlled DNA damage response (*Mashimo et al., 2013*). Here, we have described ARH3's Ser-ADP-ribosylhydrolase activity which offers a more parsimonious explanation of the susceptibility of ARH3 KO cells to DNA damage. We have previously observed DNA damage induced Ser-ADP-ribosylation of histones and other proteins (*Bonfiglio et al., 2017a*), and now we report that ARH3 is necessary for the reversal of these modifications. We also detected persistence of ADPr on numerous proteins in human ARH3 KO cells and found that ARH3 is a major hydrolyse that demodifies these proteins post DNA damage. These data suggest a conserved role for ARH3 in the response to DNA damage and the regulation of ADPr signalling.

The ADP-ribose linked to serine via an *O*-glycosidic bond is chemically similar to the ribose-ribose bond within the ADP-ribose polymer (*Bonfiglio et al., 2017b*; *Leidecker et al., 2016*). Consequently, it may be unsurprising that ARH3 can both degrade PAR chains and remove MARylation from serine residues. This observation raises a discrepancy between PARG and ARH3; both are able to hydrolyse PAR, neither are able to cleave ester bonds between the proximal ADPr unit and protein

(*Barkauskaite et al., 2013*; *Oka et al., 2006*), yet ARH3 can cleave Ser-ADPr glycosidic bonds and completely remove the ADPr chains attached to serine residues in proteins. Architecturally, both enzymes are clearly distinct from each other, with PARG adopting a macrodomain fold with a unique catalytic loop allowing for hydrolysis of ADPr, whilst ARH3 has an eponymous ARH fold featuring a binucleate $Mg^{2+}$ catalytic site (*Barkauskaite et al., 2013*; *Mueller-Dieckmann et al., 2006*; *Slade et al., 2011*). Both enzymes perform acid-base catalysis in PAR hydrolysis, but PARG has a preference for binding to the long PAR chains and is inefficient in hydrolysing short oligo-ADPr fragments and incapable or removing mono-ADPr (*Barkauskaite et al., 2013*; *Hatakeyama et al., 1986*; *Mueller-Dieckmann et al., 2006*). Altogether, these observations suggest functional differences between PARG and ARH3, but also potential redundancies. More elaborate future structural and functional studies will be needed to resolve these questions as well as the Ser-ADPr specificity of ARH3.

Advanced mass spectrometry has played an indispensable role in the discovery of Ser-ADPr as a widespread signal in DNA repair and in the identification of the enzymes and cofactors responsible for its attachment (*Bonfiglio et al., 2017a*; *Leidecker et al., 2016*). However, these findings have opened up many questions that currently cannot be addressed efficiently as only a limited number of researchers have both expertise and access to advanced instrumentation needed for unbiased and unambiguous mass spectrometric analysis of ADPr (*Bonfiglio et al., 2017b*). Thus to broaden the study of the amino acid specificities of ADPr we employed the strikingly specific activity of ARH3 as a practical analytical tool to discern the presence of Ser-ADPr without the need for mass spectrometry (*Figure 6C and D*). In combination with available biological techniques and tools, such as the anti-pan ADPr binding reagent (*Gibson et al., 2016*), we showed that ARH3 allows a simple exploration of histone Ser-ADPr dynamics during DNA damage response.

In summary, here we present ARH3 as a serine ADP-ribosylhydrolase that functions to reverse the PARP1/HPF1 mediated modification of histones and other proteins modified on serine residues. This thread of investigation has delineated a novel aspect of cellular ADP-ribosylation, an intricate level of control of ADPr signal transduction that requires specific protein tools. Understanding the activities, tendencies and regulation of these proteins may provide key insights into diseases and disorders that either rely on common ADP-ribosylation pathways, or suffer from the lack of a key protein that breaks the carefully balanced ADPr homeostasis.

## Material and methods

### Antibodies

Anti-PAR Polyclonal Antibody (rabbit) was purchased from Trevigen (Gaithersburg, MD , U S ). Pan-ADPr antibody (a macrodomain recombinant fusion protein that binds all forms of ADPr) was purchased from Millipore (Billerica, MA, US ). Rabbit anti-ARH3/ADPRHL2 (HPA027104; RRID:AB_10601330) was purchased from Atlas Antibodies. Mouse anti-GAPDH (RRID:AB_2107445) and rabbit anti-H3 (RRID:AB_417398) were purchased from Millipore.

### Plasmid construction

The relevant genes were cloned from their corresponding cDNAs and cloned into pET28 vectors unless stated otherwise. The expression construct for ARH3 was a gift from Prof Paul Hergenrother (University of Illinois). ARH1 and ARH2 were cloned into pDEST17 vector. pASK60-OmpA-mARTC2.2 6xHis-Flag tag was a gift from Prof Friedrich Koch-Nolte (Universitätsklinikum Hamburg-Eppendorf). ARH3 catalytic mutants D77N, D78N, D77N D78N were made using the QuikChange Lightning Site-Directed Mutagenesis Kit p u r c h a s e d f r o m Agilent Technologies ( S a n t a   C l a r a ,   C a , U S) as per the manufacturer's recommendations.

### Protein expression and purification and peptides

Most of the proteins were expressed and purified essentially as described (*Dunstan et al., 2012*), but with addition of a size-exclusion chromatography purification using HiLoad 16/60 Superdex 75 column. In the case of ALC1 and histone MACROH2A proteins only the macrodomains were purified rather than full length proteins. Mouse ARTC2.2 protein was purified as previously described (*Mueller-Dieckmann et al., 2002*). PARP1 wild type and the E988Q mutant were expressed and purified as previously reported (*Langelier et al., 2011*). PARG was purified as described (*Lambrecht et al.,*

2015). HPF1 was expressed and purified as reported (*Gibbs-Seymour et al., 2016*). ARH1, ARH2, ARH3 proteins were all expressed and purified using the method described (*Kernstock et al., 2006*). Histone H1 (full length) was purchased from NEB (Ipswich, MA, US), Histone H3 peptide (aa 1-21) from Sigma (Saint Louis, MO, US) and Histone H2B peptide (aa 1-21) from Millipore.

## In vitro glycohydrolase assays

Recombinant proteins or peptides were ADP-ribosylated by PARP1 in the presence or absence of HPF1 to produce substrates for the hydrolase reaction. The PARP reaction buffer contained 50 mM Tris-HCl pH 8.0, 100 mM NaCl, 2 mM $MgCl_2$, activated DNA and 50 µM $NAD^+$ spiked with $^{32}P$-$NAD^+$. The modification reaction proceeded at room temperature for 20 min before addition of the PARP inhibitor Olaparib at 1 µM. This modification reaction was used as a substrate and incubated with various glycohydrolases for 30 min at room temperature. Reactions were then analysed by SDS-PAGE and autoradiography. PARP1 concentration in the assays was 1 µM unless stated otherwise, HPF1 was always equimolar to PARP1, histone H1 was used at the final concentration of 2 µM, histone peptides were used at 0.5 µg per reaction, ARH3 and other glycohydrolase and macrodomain containing proteins were used at 3 µM unless stated otherwise.

## Ser-ADPr histone peptide purification

H3 and H2B Histone peptides were modified by PARP1 and HPF1 as described above. Subsequently, PARP1 and HPF1 were removed from the samples by filtering the reaction with a 10 kDa cut-off concentration column (Millipore). Excess of $NAD^+$ was removed using a G25 spin column (GE HealthCare, UK).

## Removal of arginine ADP-ribosylation from proteins in K652 cell extracts

In vitro modification of proteins from K562 cell extracts by ARTC2.2 recombinant protein was performed as described (*Palazzo et al., 2016*). Briefly, $8 \times 10^6$ cells were washed twice in PBS and lysed in 50 mM Tris-HCl pH 7.5, 150 mM NaCl, 0.5% Triton X-100, 0.2 mM DTT, 1 µM Olaparib, 4 mM Pefabloc SC PLUS (Sigma-Aldrich) at 4°C. The cell extract was clarified by centrifugation and diluted five times in no-Triton X-100 buffer and supplemented with 15 mM $MgCl_2$ and 1 µCi (37 kBq) of $^{32}P$-$NAD^+$. 67 µL extract aliquots were then incubated with or without 1 µM recombinant mARTC2.2 protein for 15 min at 30°C. After 15 min incubation, lysates were further incubated in presence of hydrolases for 45 min at 30°C. Samples were then analysed by SDS-PAGE and autoradiography.

## Lysine ADP-ribosylation of H3 histone peptide

Lysine ADP-ribosylation of Histone H3 peptide was performed as described (*Jankevicius et al., 2013*). Briefly, 2 µg of H3 peptide was incubated with ADP-ribose in 25 mM Tris-HCl pH 8.3 at 37°C for 3 days. Excess nucleotides were removed by purification of the peptide using ZipTip Pipette Tips (Millipore). Filter tips were pre-washed with 100% acetonitrile (ACN), and subsequently equilibrated 4 times with 0.1% trifluoroacetic acid (TFA). TFA was then added to Lys-ADP-ribosylated histone H3 to a final concentration of 0.1% and applied to the filter. The filter was then washed twice with 0.1% TFA before the histone H3 peptide was eluted with 50% ACN and 0.1% TFA. The eluate was desiccated and then resuspended in 25 mM Tris-HCl pH 8.0, 150 mM NaCl, 1 mM DTT and 2 mM $MgCl_2$. The ADP-ribosylated peptide was then analysed by the glyocohydrolase assay described above. Incubation with 0.5 M KOH at 37°C for 15 min was used as a positive control for the reaction.

## Analysis of the Ser-ADPr hydrolysis time course

Ser-ADP-ribosylated histone H3 peptide was incubated with 200 nM ARH3. At the indicated time points, samples were taken, containing 0.5 µg H3 Histone peptide, and the reactions were stopped with 1 mM ADPr and LDS sample buffer (Life Technologies, Carlsbad, CA, US) and incubated for 2 min at 90°C. The samples were analysed by SDS-PAGE and subsequent autoradiography using phosphor screen and read out with Bio-Rad Molecular Imager PharosFX Systems (Bio-Rad, Hercules, CA, US). Band intensities were quantified with the Image Lab software (Bio-Rad, version 5.0).

## Cell culture and SILAC labeling

Human U2OS osteosarcoma cells were acquired from ATCC (ATCC HTB-96; RRID:CVCL_0042), identity was confirmed by STR profiling, and absence of mycoplasma contamination confirmed by MycoAlert Mycoplasma Detection Kit. Cells were cultured in DMEM supplemented with 10% FBS and penicillin–streptomycin (100 U/ml) at 37°C, 5% $CO_2$. U2OS were regularly tested for mycoplasma by PCR-based detection analysis and discarded if positive.

For SILAC labeling (*Ong et al., 2002*), U2OS cells were grown in medium containing unlabeled L-lysine (L8662, Sigma-Aldrich) for the light condition or isotopically labeled L-lysine ($^{13}C_6,^{15}N_2$, 608041, Sigma-Aldrich) for the heavy condition. Both light and heavy DMEM were supplemented with 10% dialyzed FBS (Thermo Scientific). Cells were cultured for more than seven generations to achieve complete labeling. Incorporation efficiency (>99%) was determined by MS.

For the generation of ARH3 KO cell lines, a pair of sgRNA were designed to generate deletion into the ARH3 gene. More specifically, sgRNA 210 (GCGCTGCTCGGGGACTGCGT) and sgRNA 212 (GGGCGAGACGTCTATAAGGC), were used to remove part of ARH3 exon 1 and part of the downstream intron, including the splice donor site. sgRNAs were cloned into epX459(1.1), which Dr Joey Riepsaame, (GEO facility, Sir William Dunn of Pathology, Oxford) generated by subcloning enhanced Cas9 (eSpCas9) v1.1 into plasmid pX459.

U2OS cells were transfected with control sgRNA or co-transfected with sgRNAs 210 and 212 (1:1 ratio) using TransIT-LT1 Transfection Reagent (Mirus), following the manufacturer's instructions. After 24 hr, transfected cells were treated with 1 µg/mL Puromycin (InvivoGen) for 36 hr. After selection, control and ARH3 KO cells were diluted to 0.4 cells/100 µl and seeded on 96-well plates. Single colonies were amplified and screened by western blot using anti-ARH3 antibody (HPA027104 Atlas Antibodies). Colonies containing Flag-tagged Cas9 were discarded.

## Preparation of cell extracts

U2OS cells were stimulated with 2 mM $H_2O_2$ for the indicated time points and lysed in the following buffer: 50 mM Tris-HCl pH 8.0, 100 mM NaCl, 1% Triton X-100. Immediately before lysing the cells, the lysis buffer was supplemented with 5 mM $MgCl_2$, 1 mM DTT, proteases and phosphatases inhibitors (Roche), 1 µM ADP-HPD, and 1 µM Olaparib. After the cell pellet was re-suspended in the supplemented lysis buffer, Benzonase (Sigma) was added.

## In vitro complementation of ARH3-KO cell extracts

ARH3 KO cells from 10 and 120 min post $H_2O_2$ time points were lysed as previously described and diluted 1:5 in PARP reaction buffer supplemented with 1 µM Olaparib. Aliquots of diluted extract were incubated for 1 hr at room temperature with buffer or 0.5 µM ARH3, MACROD2 and PARG. Reactions were stopped with loading buffer and subjected to a standard SDS-PAGE method.

## Histone purification

SILAC-labeled U2OS cells were stimulated with 2 mM $H_2O_2$ for the indicated time points and core histones were purified as previously described (*Leidecker et al., 2016*). Briefly, cells were washed twice with ice-cold PBS and lysed by rotation in 0.1 M $H_2SO_4$ at 4°C for 2 hr. The lysate was centrifuged at 2200 g at 4°C for 20 min. The pellet with non-soluble proteins and cell debris was discarded. Sulfuric acid-soluble proteins were neutralized with 1 M Tris-HCl pH 8.0. NaCl, EDTA and DTT were added to a final concentration of 0.5 M, 2 mM, 1 mM, respectively. For ion exchange chromatography, sulfopropyl (SP)-Sepharose resin was packed into a column and pre-equilibrated with 10 volumes of binding buffer (50 mM Tris-HCl pH 8.0, 0.5 M NaCl, 2 mM EDTA, 1 mM DTT). The neutralized supernatant containing $H_2SO_4$-soluble proteins was passed through the column. The resin was washed with 10 volumes of binding buffer and 30 volumes of Washing Buffer (50 mM Tris-HCl pH 8.0, 0.6 M NaCl, 2 mM EDTA, 1 mM DTT). Proteins were eluted with elution buffer (50 mM Tris-HCl pH 8.0, 2 M NaCl, 2 mM EDTA, 1 mM DTT) in 10 fractions. Eluted proteins (mainly core histones) were precipitated overnight in 4% (v/v) PCA at 4°C. The fractions were then centrifuged at 21,000 g at 4°C for 45 min and the resulting pellets were washed with 4% PCA (2 × 1 ml), 0.2% HCl in acetone (2 × 1 ml), acetone (2 × 1 ml).

## Western blot analysis

For Western blot analysis, samples were subjected to a standard SDS-PAGE method. Proteins were transferred to PVDF membranes (Merck Millipore). Membranes were then blocked with TBS-T buffer (25 mM Tris-HCl pH 7.5, 150 mM NaCl, 0.05% Tween 20% and 5% non-fat dried milk) and incubated overnight with primary antibodies at 4°C, followed by a one hour incubation with peroxidase-conjugated secondary antibodies at room temperature. Blots were developed using ECL Select and signals were captured using a ChemiDoc MP System (Bio-Rad). Dilutions used for the primary antibodies were: Anti-poly-ADP-ribose: diluted at 1:2000, Anti-pan-ADP-ribose binding reagent: diluted 1:1000.

## Mass spectrometric analysis

Proteins were digested using partial FASP as previously described (*Leidecker et al., 2016*). After digestion, peptides were desalted on either C18 cartridges (3M Empore) or using in-house manufactured StageTips (*Rappsilber et al., 2003*), depending on the peptide amounts. Eluted peptides were dried down in a Speedvac concentrator and resuspended in 0.1% FA prior to LC-MS/MS analysis. Liquid chromatography and MS for all LC-MS/MS runs were performed as previously described (*Bonfiglio et al., 2017a*). For unambiguous identification of ADPr sites on PARP1 E988Q and H1 from in vitro reactions (*Figure 2—figure supplement 1*), ETD fragmentation was used, with the same parameters as previously described (*Bonfiglio et al., 2017a*). For histone Ser-ADPr quantification (*Figure 1D and E*, and *Figure 1—figure supplement 1*) HCD fragmentation was used, using the same conditions as previously described (*Bonfiglio et al., 2017a*). Raw files were analyzed with MaxQuant proteomics suite of algorithms (version 1.5.3.17) (*Cox and Mann, 2008*), using the same parameters as previously described (*Bonfiglio et al., 2017a*).

For *Figure 1C*, normalized SILAC ratios from ADPr(S) sites table were extracted from MaxQuant, $\log_2$ converted and relativized to the non-treated condition to create the scatterplots comparing the relative abundance versus the time of 2 mM $H_2O_2$. For *Figures 1D* and *5D* and *Figure 1—figure supplement 1A* representative MS spectra were manually selected and annotated. For *Figure 2D* and *Figure 1—figure supplement 1B* evidence tables from MaxQuant were analyzed using Perseus software (http://www.perseus-framework.org) together with an in-house script (*Source code 1.*) to create the scatterplots comparing SILAC ratios versus peptide intensities (*Bonfiglio et al., 2017a*).

## Acknowledgements

Ahel laboratory is funded by the Wellcome Trust (grant 101794), Cancer Research UK (grant C35050/A22284), and the European Research Council (grant 281739). The work in Matic laboratory was funded by the Deutsche Forschungsgemeinschaft (Cellular Stress Responses in Aging-Associated Diseases) (grant EXC 229 to IM) and the European Union's Horizon 2020 research and innovation program (Marie Skłodowska-Curie grant agreement 657501 to JJB and IM). We are grateful to Paul Hergenrother and Friedrich Koch-Nolte for providing the protein expression constructs. Thanks to Qi Zhang (Max Planck Institute for Biology of Ageing) for helping with some experiments. We are grateful to Gytis Jankevicius and Dragana Ahel for the gift of proteins and Joey Riepsaame for the CRISPR design (Sir William Dunn School of Pathology).

## Additional information

### Funding

| Funder | Grant reference number | Author |
|---|---|---|
| H2020 Marie Skłodowska-Curie Actions | 657501 | Juan José Bonfiglio Ivan Matic |
| Deutsche Forschungsgemeinschaft | EXC 229 | Ivan Matic |
| Wellcome | 101794 | Ivan Ahel |
| Cancer Research UK | C35050/A22284 | Ivan Ahel |
| European Research Council | 281739 | Ivan Ahel |

The funders had no role in study design, data collection and interpretation, or the decision to submit the work for publication.

### Author contributions
PF, JJB, LP, EB, Investigation, Formal analysis, Writing—original draft, Writing—review and editing ; IM, Investigation, Formal analysis, Writing—original draft, Writing—review and editing, Supervision, Funding acquisition; IA, Formal analysis, Supervision, Funding acquisition, Writing—original draft, Writing—review and editing

### Author ORCIDs
Juan José Bonfiglio, http://orcid.org/0000-0001-7767-0799
Luca Palazzo, http://orcid.org/0000-0002-5556-5549
Ivan Matic, http://orcid.org/0000-0003-0170-7991
Ivan Ahel, http://orcid.org/0000-0002-9446-3756

## Additional files

### Supplementary files
• Source code 1. R-script used to analyse the MaxQuant data.

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
