## [Decision Letter]

Thank you for submitting your article "Serine ADP-ribosylation removal by the hydrolase ARH3" for consideration by *eLife*. Your article has been reviewed by three peer reviewers, and the evaluation has been overseen by Ivan Dikic as Reviewing Editor and Senior Editor. The reviewers have opted to remain anonymous.

The reviewers have discussed the reviews with one another and the Reviewing Editor has drafted this decision to help you prepare a revised submission.

Summary:

The authors present a thorough analysis of the reversal of the posttranslational modification of serines with ADP-ribose. They report the identification of ARH3 as the ADP-ribosylhydrolase working on Ser-ADP-ribosylation. The study used mass spectrometry to clearly demonstrate that the serine modification is reversible. Using a panel of purified proteins that were considered likely candidates to process serine-ADP-ribose, the study identified ARH3 as an enzyme capable of reversing this modification. The panel of enzymes make a nice tool set for the identification of potential modification types in other proteins. The identification of serine modification is an exciting development in the PARP field, and the current study addresses a key question that many groups are certainly working toward.

With the identification of ARH3 as the serine-ADP-ribosylhydrolase, serine ADP-ribosylation is now also an enzymatically fully reversible post-translational modification. This is a very important finding and warrants high visibility that would be achieved by publishing in *eLife*.

Essential revisions:

1) The relevance of ARH3 in reversing Ser-ADPrylation in vivo. Authors identify ARH3 as an enzyme that efficiently reverses serine ADPrylation in vitro. The presented experiments do not exclude the possibility that other yet unknown enzymes might also reverse serine ADPrylation and that ARH3 is important for the reversal in a cellular context. The authors should test whether ARH3 is required for removal of SerADPr in vivo, for instance by knocking down ARH3 and then monitoring the amount of Serine-ADPr by mass spectrometry and WB.

2) Specificity: The biochemical assays showed virtually complete reversal of histone peptide serine-ADP-ribose in the presence of ARH3, yet the signal for modified PARP1 protein was reduced to a similar level by PARG and ARH3. Why is ARH3 not efficient in completely removing ADPr from PARP1 whereas it does so on histones (Figure 1). These reactions were done with HPF so the ADPr on PARP1 should be on serine. Could the authors provide some information about the activity of PARG and ARH3 in removing poly SerADPr? Is one enzyme more active than another in assays that are done by authors? What is to be expected in vivo?

3) The manuscript could be improved by giving a better description of what we already know about ARH3 in terms of cellular localization, expression level, etc., The discussion about the possible importance of ARH3 in DNA damage response in light of previous studies on ARH3 and findings presented in this manuscript should be extended.

---

## [Author Response]

Essential revisions:

*1) The relevance of ARH3 in reversing Ser-ADPrylation* in vivo*. Authors identify ARH3 as an enzyme that efficiently reverses serine ADPrylation* in vitro*. The presented experiments do not exclude the possibility that other yet unknown enzymes might also reverse serine ADPrylation and that ARH3 is important for the reversal in a cellular context. The authors should test whether ARH3 is required for removal of SerADPr* in vivo*, for instance by knocking down ARH3 and then monitoring the amount of Serine-ADPr by mass spectrometry and WB.*

We have now included new data showing a major role for ARH3 in removal of Ser-ADPr in human cells (Figure 5). We generated ARH3 KO cells in which we show by WB and MS that levels of Ser-ADPr are drastically increased when compared to control cells, especially post DNA damage. MS analyses showed that ARH3 KO cells show strong increase in Ser-ADPr of histones (40-fold for the best quantified histone ADPr mark, H2BS6-ADPr) 2 hours post DNA damage (new Figure 5). Finally, in vitro complementation experiments with recombinant enzymes (Figure 5) show that ADP-ribosylation persisting in ARH3 KO cell extracts can be efficiently erased by adding recombinant ARH3, but not by PARG or MACROD2.

*2) Specificity: The biochemical assays showed virtually complete reversal of histone peptide serine-ADP-ribose in the presence of ARH3, yet the signal for modified PARP1 protein was reduced to a similar level by PARG and ARH3. Why is ARH3 not efficient in completely removing ADPr from PARP1 whereas it does so on histones (Figure 1). These reactions were done with HPF so the ADPr on PARP1 should be on serine. Could the authors provide some information about the activity of PARG and ARH3 in removing poly SerADPr? Is one enzyme more active than another in assays that are done by authors? What is to be expected* in vivo*?*

We suspect that this comment addresses Figure 2 and C. The in vitro activity of ARH3 against PAR on PARP1 chains is 10 times slower than that of PARG (Mueller-Dieckmann et al., 2006; Oka, Kato and Moss, 2005). In our in vitro reactions, we also have substrates with serine-ADPr that additionally compete with modified PARP1 for ARH3 activity. As a consequence, ARH3 cannot fully process the PAR substrate and often yields a smeared PAR signal, rather than neatly collapsing the PAR signal into a single band as always seen with PARG (see for example the new Figure 2—figure supplement 1). We further used a combination of PARG and ARH3 to show that the majority of PARylated sites on PARP1 (in the presence of HPF1) are on serine residues (Figure 2—figure supplement 1).

Our new data shows that the situation is similar in cells (new Figure 5); while dynamics of the long PAR chains is less affected in ARH3 KO cells thanks to active PARG protein (Figure 5; panel with PAR antibody), the removal of short oligo- and mono-ADPr chains is dramatically affected in ARH3 KO cells (Figure 5, panel with pan-ADPr antibody). These observation match the in vitro data and the model proposed by Hatakeyama et al., 1986 which shows that PARG preferentially cleaves long PAR chains, with the *K*_m_ value for long PAR chains being 1% of that for small ones.

*3) The manuscript could be improved by giving a better description of what we already know about ARH3 in terms of cellular localization, expression level, etc., The discussion about the possible importance of ARH3 in DNA damage response in light of previous studies on ARH3 and findings presented in this manuscript should be extended.*

The knowledge about the cellular distribution of ARH3 and phenotypes induced by its loss of function are very limited (Niere et al., 2012; Mashimo et al., 2013). We have now extended the discussion, reporting all that is currently known.